# COVID-19 Vaccine Failure in a Patient with Multiple Sclerosis on Ocrelizumab

**DOI:** 10.3390/vaccines9030219

**Published:** 2021-03-04

**Authors:** Sridhar Chilimuri, Nikhitha Mantri, Sudharsan Gongati, Maleeha Zahid, Haozhe Sun

**Affiliations:** BronxCare Health System, Department of Medicine, Affiliated with Icahn School of Medicine at Mount Sinai, Bronx, NY 10457, USA; chilimuri@bronxcare.org (S.C.); nmantri@bronxcare.org (N.M.); sgongati@bronxcare.org (S.G.); mzahid@bronxcare.org (M.Z.)

**Keywords:** COVID-19, vaccination, B-Cell, ocrelizumab, multiple sclerosis, humoral immunity

## Abstract

Vaccines will play a key role in ending the COVID-19 pandemic. Vaccination against infections remains an important part of the management of patients with multiple sclerosis. However, there are limited data about the safety and efficacy of the currently available COVID-19 mRNA vaccines in patients with multiple sclerosis receiving concurrent immunosuppressive therapies. Patients on B cell depleting therapy such as ocrelizumab have an attenuated vaccine response. We report the first case of COVID-19 vaccine failure in a patient with relapsing-remitting multiple sclerosis on B cell depleting therapy, ocrelizumab. We offer suggestions to improve vaccine efficacy in these patients.

## 1. Introduction

The Pfizer-BioNTech (Brooklyn NY USA) and Moderna (Cambridge MA USA) COVID-19 mRNA vaccines received emergency use authorization by the US FDA in December 2020. As of 11 February 2020, more than 68 million doses of COVID-19 vaccines have been administered [1]. There are limited data about the safety and efficacy of the currently available COVID-19 mRNA vaccines in patients receiving concurrent immunosuppressive therapies. In this report, we present a case of COVID-19 vaccine failure due to the concurrent use of ocrelizumab, a disease-modifying therapy with B cell-depleting effects, used in the treatment of primary and secondary progressive multiple sclerosis.

## 2. Case

Our patient is a 52-year-old Caucasian male with a history of hypertension and multiple sclerosis. He resides in New York City and works in a profession that makes him at high risk for exposure to the severe acute respiratory syndrome coronavirus 2 (SARS-CoV-2). He takes losartan 100 mg and hydrochlorothiazide 12.5 mg daily for his hypertension. He was diagnosed with the relapsing-remitting form of multiple sclerosis 15 years ago and was initially treated with interferon beta-1a and also received intravenous immunoglobulins for a brief period of time. His last relapse was six years ago. Two years ago, he was started on ocrelizumab and received his last infusion in the first week of December 2020 Subsequently, when the COVID-19 vaccines were made available, he received both doses of the Pfizer-BioNTech COVID-19 vaccine on 19 December 2020, and 12 January 2021.

On 31 January 2021, approximately 19 days after receiving the last dose of the COVID-19 vaccine, he started experiencing generalized malaise, myalgias, and a mild cough. He tested positive for COVID-19 via a reverse transcription polymerase chain reaction (RT-PCR) nasopharyngeal swab. Serological testing was performed on day 4 of symptoms onset, with two separate assays, assessing the immunological response to the spike and neuclocapsid protein of SARS-CoV-2, respectively. The VITROS COVID-19 assay was significant for positive IgM and negative IgG to the Spike (S1) protein of the SARS-CoV-2 virus. The Roche Cobas Elecsys Anti-SARS-CoV2 test was negative for both IgG and IgM to the nucleocapsid (N) antigen. This indicates recent exposure to SARS-CoV-2 and not seroconversion due to prior infection or vaccination. On day 4 of his symptoms, he received an infusion of casirivimab and imdevimab, a monoclonal antibody cocktail against SARS-CoV-2. His symptoms subsided following the infusion.

## 3. Discussion

The U.S. Food and Drug Administration (FDA) approved ocrelizumab on 28 March 2017for the treatment of adult patients with relapsing or primary progressive forms of multiple sclerosis. As of December 2020, more than 200,000 patients with multiple sclerosis have initiated ocrelizumab therapy globally as part of clinical trials and post marketing experience, amounting to a total of >300,000 patient-years [2].

The disease course of multiple sclerosis is thought to be influenced by B-cells, through mechanisms such as antigen presentation, autoantibody production, cytokine regulation, and formation of ectopic lymphoid aggregates in the meninges [3]. The exact mechanism by which ocrelizumab exerts its therapeutic effects in multiple sclerosis is presumed to involve binding to CD20 on pre-B and mature B lymphocytes. This results in a B cell-depleting effect via antibody-dependent cell-mediated phagocytosis and cytotoxicity, as well as complement-mediated lysis [3].

In a recently published, randomized, open-label, Phase IIIb A Phase 3 trial, VELOCE (NCT02545868), treatment with ocrelizumab was linked to an attenuated humoral immune response to the tetanus, seasonal flu, and pneumococcus vaccines in patients with relapsing multiple sclerosis [4]. CD19 levels were effectively depleted within two weeks of ocrelizumab infusion and remain depleted up to 6 months or longer in the majority of patients in that study. However, these patients can still mount humoral responses to multiple vaccines, albeit reduced, when vaccinations were administered three months after the patients had received ocrelizumab [4].

Vaccination against infections remains an important part of the management of patients with multiple sclerosis. In order to achieve maximal vaccine efficacy, the timing of COVID-19 vaccination remains a key consideration, especially in patients with multiple sclerosis on B cell-depleting therapy. We suggest timing the vaccination within a six-week window where the immunosuppressive effects of such therapy would be at their lowest (e.g., dosing at the end of an infusion cycle before the next infusion) while giving ample time for the vaccines to reach their peak efficacy [5,6]. However, this may not always be possible, and vaccination should not be deferred.

Severe COVID-19 infection is more likely in patients with multiple sclerosis who are older, have a higher baseline Expanded Disability Severity Scale (EDSS) score, have comorbidities, or are receiving B cell depleting therapy [7]. Our patient was at a high risk of occupational exposure, and vaccinating such individuals is of paramount importance. For such high-risk patients, the benefits of early vaccination in preventing a COVID-19 infection would still outweigh the possible risks of vaccination failure. 

## 4. Conclusions

This is the first report of vaccine failure in a patient with a long-standing history of relapsing-remitting multiple sclerosis on ocrelizumab. Our findings are in line with the recent clinical trial findings described above. Importantly, our case provides further evidence and broadens the recent discussion on developing effective COVID-19 vaccination strategies, such as dose interruption, in patients receiving concurrent B cell depleting therapy [8]. Although there are multiple types of vaccines currently under development, live and live-attenuated vaccines are not recommended during ocrelizumab treatment and until B-cell repletion, further limiting the vaccine choices in these patients. Finding an optimal interval for vaccination in patients on ocrelizumab may improve vaccine efficacy.

## Data Availability

The data presented in this study are available on request from the corresponding author.

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
