# Peer review of "COVID-19 Vaccine Failure in a Patient with Multiple Sclerosis on Ocrelizumab"

_vaccines, 2021, doi:10.3390/vaccines9030219_

Round 1

Reviewer 1 Report

Authors described a case report about failure of COVID-19 vaccine in a multiple sclerosis patient who received anti-CD20 monoclonal antibody. Goal of vaccination is to induce acquired immunity including humoral and cell-mediated immunity. Especially, the humoral immunity is important to prevent host cells from infection . I understand vaccination does not work if B-cells are depleted from a host. I agree the case report has obvious but important information about vaccination strategy to MS patients. Authors should show percentages of CD19-positive cells in blood of the patient before and after treatment with Ocrelizumab if they have.

Author Response

We would like to thank the reviewers for their invaluable comments. Unfortunately, we do not have CD19 levels of this reported individual. However, in our future clinical studies, obtaining CD19+ cells will be one of our priorities.

Reviewer 2 Report

I was invited to revise the paper entitled "COVID-19 Vaccine Failure in a Patient with Multiple Sclerosis on Ocrelizumab". It was a very interesting case report describing the low efficacy of SARS-Cov2 vaccine in a patient with multiple sclerosis treated with Ocrelizumab. It focused on an important topic and can improve knowledge in this field, allowing the implementation of new observational studies on this kind of patient.

I have only some minor observations:

  • In Introduction section Authors should describe the action of Ocrelizumab. Readers of our journal are not all Nueologists/Haematologists.
  • In discussion section Authors should report previous recent study on LB monoclonal antibody. For example this recent review  doi: 10.1111/cei.13495. Epub 2020 Aug 1.or this paper  doi: 10.1007/s00415-020-10045-9. Epub 2020 Jul 7.

Author Response

We would like to thank the reviewer for the invaluable comments. The following are our amendments based on the provided suggestions.

  • We have included a short introduction to the indication of Ocrelizumab in MS patients in lines 23-26.
  • Additionally, we have expanded on our description of the mechanism of action of Ocrelizumab in Lines 57 - 63.
  • Finally, we have referenced Baker et al's review in Lines 87-91.

Best Regards,

Haozhe Sun